# The association between *Toxoplasma gondii* infection and asthma in the United States: A cross-sectional survey analysis

Heather Anholt *

School of Population and Public Health, Faculty of Medicine, University of British Columbia, Vancouver, British Columbia, Canada

* hezyvet@gmail.com

## Abstract

The hygiene hypothesis proposes that declining exposure to microbial influences early in life is implicated in the rising trend of allergy and asthma in high-income societies. Approximately 8% of Americans have been diagnosed with asthma, representing 25 million people, and understanding how the human microbiome affects asthma could help guide exposure recommendations or microbe-based therapeutics. *Toxoplasma gondii* is a common gastro-intestinal microorganism that may modulate immune function. We used a cross-sectional study design to examine a public database of U.S. residents aged 6–80 years or older from the 2012–2014 survey cycles of the American National Health and Nutrition Examination Survey (NHANES) to construct an ordinal logistic regression model of the relationship between *T. gondii* infection and asthma. Of the 12,620 subjects tested for *T. gondii* infection, 89.2% were seronegative and 10.8% seropositive. No asthma was reported by 83.5% of subjects, while 16.5% reported varying degrees of asthma severity. We detected no significant association between *T. gondii* infection and asthma. While the unadjusted regression model suggested a small protective effect of *T. gondii* on asthma (OR = 0.90; 95% CI = 0.83–0.97), no effect was detected when the model was adjusted for key demographic factors (OR = 1.00, 95% CI = 0.91–1.10). While *T. gondii* may be a marker for the protective effect of exposure to a diversity of microbial organisms early in life, it has no apparent causal effect on asthma, or it may not be significant when considered in isolation.

## Introduction

Asthma is a complex syndrome characterized by increased bronchial reactivity and airway obstruction that can result in death [1]. Approximately 8% of Americans have been diagnosed with asthma, representing 25 million people (1–2). Between 1980

---

**Data availability statement:** All data files are available from the 2012-2014 survey cycles of the American National Health and Nutrition Examination Survey (NHANES) and are available from https://wwwn.cdc.gov/nchs/nhanes/.

**Funding:** The author(s) received no specific funding for this work.

**Competing interests:** The authors have declared that no competing interests exist.

and 2011, the prevalence of asthma in the United States (U.S.) more than doubled and similar trends have been observed in other highly industrialized nations [2–4].

The hygiene hypothesis proposes that declining exposure to microbial influences early in life is implicated in the rising trend of allergy and asthma in high-income societies. Children who grow up on a farm are at lower risk for asthma and asthma risk decreases as the number of older siblings increases, possibly due to increased exposure to microorganisms and their products through contact with older siblings [2,5]. Understanding how the human microbiome is shaped and affects health can provide mechanistic explanation to the hygiene hypothesis, which could help guide exposure recommendations or microbe-based therapeutics for disease prevention and modification [5,6].

*Toxoplasma gondii* is an obligate intracellular gastro-intestinal protozoan, and nearly ubiquitous microorganism, whose definitive host is cats. People become infected with *T. gondii* through indirect fecal-oral transmission or consumption of undercooked meat. About 12% of the U.S. population is infected, and in some countries, prevalence is as high as 60% [7,8]. Infection with *T. gondii* induces a strong cell-mediated immune response, with the production of IFN-g cytokines by polarized T helper type 1 cells (Th1) in the early stages of infection [9,10]. Conversely, the pathophysiology of asthma is characterized by the development of hyperreactivity due to cytokines produced by T helper type 2 cells (Th2). Infection with *T. gondii* may exert a protective effect on the development of antigen-induced airway inflammation through high concentrations of Th1 cytokines and a reduction of allergen-specific Th2-associated cytokines associated with the *T. gondii* immune response [11]. Indeed, Fenoy *et al.* (2009) showed that *T. gondii* infection substantively blocked the development of antigen-induced airway inflammation in adult laboratory (BAL-B/c) mice [9]. Despite recognized impacts on the mammalian immune response, however, the relationship between *T. gondii* infection and asthma has not been well investigated [6,8]. In the current study, we quantify the association between *T. gondii* infection and asthma in the U.S. population.

## Methods

We used a cross-sectional study design to examine a public database of U.S. residents aged 6–80 years or older from the 2012–2014 survey cycles of the American National Health and Nutrition Examination Survey (NHANES). This is a continuous survey based on a stratified multistage clustered probability design to select a representative sample of the civilian, noninstitutionalized U.S. population [12]. Identification of individuals is not possible from these data. Only subjects 6 years of age or older were tested for *T. gondii*. Demographic, socioeconomic, and health-related information were taken from the NHANES Questionnaire, and serological data from the NHANES surplus sera program (2012–2014). The data were accessed on October 16th, 2023.

The University of British Columbia's Policy LR9, item 7.10.3, on studies involving human participants and Article 2.2 of the Tri-Council Policy Statement on Ethical Conduct for Research Involving Humans provide ethical support for this study [13,14].

The NHANES protocols were approved by the National Center for Health Statistics institutional review board, and written informed consent was obtained from all participants [12].

The response rates for in-home interviews in 2011–2012 and 2013–2014 were 72.6 and 71.0%, respectively [12]. Of the 19,931 persons interviewed, 16,733 (84%) were ≥ 6 years old. Of these 16,733 interviewed subjects, 12,677 (76%) had sera tested for *T. gondii* antibodies. Three of these subjects were excluded due to missing IgG results, and an additional 45 were excluded because they most likely had acute infections (exposure occurred within the last two weeks); the cross-sectional NHANES data would probably not capture the effects of an acute infection [15]. Nine subjects were excluded because the outcome of asthma was missing, resulting in an analytic sample of n = 12,620 (Fig 1, Table 1).

The exposure variable was *T. gondii* infection, defined as having IgG antibodies to *T. gondii* above ≥33 IU/mL. IgG antibodies were measured with the Toxoplasma IgG Enzyme Immunoassay (Bio-Rad, Redmond, WA) and IgM antibodies with the Platelia Toxo IgM (Bio-Rad). Test results were expressed as international units (IU/mL) or sample ratios, respectively. If IgG assay results were missing, the subject was excluded from the analysis (n = 3). If IgG was positive and IgM was missing, the exposure was counted as "positive". If IgG was negative (<33 IU/mL), and IgM was missing, the exposure was counted as "negative". If IgG was negative, and IgM was positive (Sample Ratio ≥1) the exposure was categorized as "acute" and excluded from the analysis (n = 45) (Fig 1) [15].

We created a proxy variable representing five levels of asthma severity from five variables in the Questionnaire. Level 0 indicates "never been told you have asthma", level 1, "told you have asthma", level 2, "told you have asthma and still have

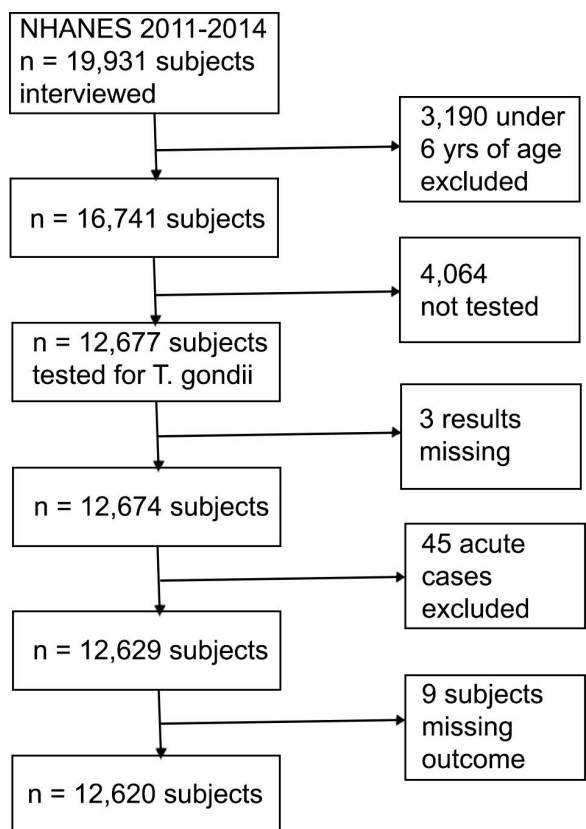

**Fig 1. Flowchart representing the process for selecting the analytic sample from the NHANES 2012-2014 survey cycle data to evaluate the association between *Toxoplasma gondii* infection and asthma in the U.S.**

**Table 1. Sample characteristics (frequencies and percentages) of individuals from the NHANES 2011-2014 survey cycles and their *Toxoplasma gondii* infection status, stratified by the outcome self-reported asthma severity (0 = "never been told you have asthma"; 1 = "told you have asthma"; 2 = "told you have asthma and still have asthma"; 3 = "still have asthma and had an asthma attack in the past year"; and 4 = "had an emergency care visit for asthma in the past year). Rao-Scott Chi-squared tested for differences between asthma severity and the identified confounders; statistically significant p-values (p < 0.05) are bolded.**

| | | | Asthma severity | | | | | |
|---|---|---|---|---|---|---|---|---|
| | Overall | 0 | 1 | 2 | 3 | 4 | |
| n | 12620 | 10544 | 839 | 615 | 424 | 198 | p-test[b] |
| *T. gondii* | n (%)[a] | n (%)[a] | n (%)[a] | n (%)[a] | n (%)[a] | n (%)[a] | 0.140 |
| Positive | 1572 (10.8) | 1365 (11.2) | 77 (6.5) | 72 (12.6) | 43 (10.4) | 15 (6.3) | |
| Negative | 11057(89.2) | 9179 (88.8) | 762(93.5) | 543(87.4) | 381(89.6) | 183(93.7) | |
| **Gender** | (%) | (%) | (%) | (%) | (%) | (%) | **0.036** |
| female | 6416 (51.2) | 5299 (50.5) | 398(53.1) | 334(51.3) | 260(54.6) | 125(75.1) | |
| male | 48.8 | 49.5 | 46.9 | 48.7 | 45.4 | 24.9 | |
| **Age** | n (%) | n (%) | n (%) | n (%) | n (%) | n (%) | **0.019** |
| 6-9 | 1058 (4.8) | 875 (4.8) | 53 (3.2) | 52 (6.4) | 43 (4.9) | 35 (12.0) | |
| 10-19 | 2508 (14.0) | 1984 (12.6) | 213(21.4) | 172(18.7) | 92 (20.2) | 47 (25.5) | |
| 20-29 | 1512(14.6) | 1217(13.8) | 150(19.8) | 67 (14.0) | 56 (21.6) | 22 (17.1) | |
| 30-39 | 1571 (14.2) | 1318(14.5) | 125(15.8) | 63(12.0) | 37 (8.3) | 28 (12.5) | |
| 40-49 | 1527 (15.2) | 1317 (15.6) | 81 (15.8) | 66 (12.1) | 43 (10.1) | 20 (12.1) | |
| 50-59 | 1508 (15.8) | 1304 (16.4) | 70 (9.7) | 59 (15.3) | 61 (18.2) | 14 (5.3) | |
| 60-69 | 1495 (11.8) | 1266 (11.9) | 89 (10.1) | 68 (13.5) | 53 (12.8) | 19 (12.5) | |
| 70-79 | 880 (6.1) | 758 (6.5) | 41 (3.9) | 45 (5.5) | 26 (2.3) | 10 (2.4) | |
| 80+ | 561 (3.5) | 505 (3.9) | 17 (0.4) | 23 (2.5) | 13 (1.7) | 3 (0.6) | |
| **Poverty index[c] (mean (SD))** | 2.85 (1.68) | 2.89 (1.67) | 2.82(1.70) | 2.60(1.78) | 2.73(1.68) | 1.86(1.38) | **0.003** |
| **Education** | n (%) | n (%) | n (%) | n (%) | n (%) | n (%) | 0.132 |
| <9th | 1145 (6.8) | 1008 (7.2) | 59 (6.6) | 44 (4.4) | 25 (4.3) | 9 (3.7) | |
| 9–11th | 1727 (11.5) | 1426 (11) | 128 (14.6) | 96 (14.7) | 44 (9.3) | 33 (16.5) | |
| high school | 2686 (18.1) | 2242 (18.2) | 162 (17.6) | 140 (20.8) | 88 (12.4) | 54 (25.3) | |
| somecollege | 3626 (31.7) | 2980 (32.0) | 247 (27.4) | 181 (27.3) | 153 (35.2) | 65 (43.2) | |
| college | 3021 (31.8) | 2549 (31.6) | 206 (33.8) | 134 (32.8) | 102 (38.8) | 30 (11.4) | |
| **# Children** | n (%) | n (%) | n (%) | n (%) | n (%) | n (%) | 0.143 |
| 0 | 6471 (59.4) | 5459 (60.1) | 415 (54.9) | 314 (61.1) | 210 (58.5) | 73 (34.8) | |
| 1 | 2411 (17.0) | 1985 (16.2) | 175 (20.6) | 112 (14.6) | 97 (24.4) | 42 (33.3) | |
| 2 | 2208 (15.2) | 1825 (15.3) | 152 (15.3) | 117 (16.5) | 68 (9.8) | 46 (19.9) | |
| 3 | 1045 (6.2) | 879 (6.1) | 62 (7.1) | 47 (6.5) | 33 (6.7) | 24 (9.8) | |
| 4+ | 485 (2.2) | 396 (2.3) | 35 (2.0) | 25 (1.3) | 16 (0.6) | 13 (2.2) | |
| **Race** | n (%) | n (%) | n (%) | n (%) | n (%) | n (%) | **0.024** |
| Hispanic | 3167 (15.3) | 2721 (15.8) | 194 (14.6) | 133 (11.6) | 68 (8.1) | 51 (25.1) | |
| Black | 2768 (10.2) | 2221 (9.9) | 192 (10.7) | 182 (14.7) | 100 (9.9) | 73 (17.1) | |
| White | 4778 (67.3) | 3944 (66.9) | 339 (68.9) | 247 (68.7) | 194 (77.2) | 54 (51.2) | |
| Asian | 1433 (4.3) | 1292 (4.6) | 67 (2.3) | 35 (3.0) | 30 (2.2) | 9 (2.4) | |
| other | 474 (2.9) | 366 (2.8) | 47 (3.5) | 18 (2.0) | 32 (2.5) | 11 (4.2) | |
| **Country of birth** | n (%) | n (%) | n (%) | n (%) | n (%) | n (%) | 0.065 |
| other | 3008 (14.8) | 2727 (15.6) | 127 (11.0) | 78 (9.6) | 48 (10.3) | 28 (13.5) | |
| U.S. | 9612 (85.2) | 7817 (84.4) | 712 (89.0) | 537 (90.4) | 376 (89.7) | 170 (86.5) | |

Data are n (%) unless otherwise stated.

[a]Percentages are calculated on complete case data, accounting for complex survey design and survey weights.

[b]p-values are calculated using Rao-Scott Chi-squared test.

[c]Poverty index of ≥3.5 indicates a high level of poverty.

asthma", level 3 "still have asthma and had an asthma attack in the past year" and level 4 "had an emergency care visit for asthma in the past year". For comparison, we created a binary outcome variable for asthma based on the question, "Has a doctor or other health professional ever told you that you have asthma?" "Yes" or "No".

We selected the minimum sufficient adjusted set of confounders using the modified disjunctive cause criterion, informed by evidence in the current literature and a directed acyclic graph (DAG) (Fig 2) [1,7,16–20]. As cat ownership is not an important risk factor for *T. gondii* infection, and pet ownership information was not collected in the 2012–2014 NHANES, cat ownership was not included as a covariate [21–26]. Because the NHANES categorizes subjects over the age of 80 years as "80+", we grouped the age variable into decades as follows: "6-9"; "10-19"; "20-29"; "30-39"; "40-49"; "50-59"; "60-69"; "70-79"; and "80+". This grouping is consistent with other analyses of *T. gondii* infection and is a clinically relevant interval for assessing trends in asthma severity and *T. gondii* seropositivity [7,27,28].

Gender was "male" or "female". Race was categorized as "Hispanic", "White", "Black", "Asian", or "other" according to the respondent's self-assessment. Country of birth was "U.S." or "other". The education level of the household reference person was categorized as "<9th grade", "9th-11th grade", "high school", "some college", or "college completed". Economic status was defined according to the poverty index of the household reference person and expressed as a continuous numeric variable (0–5), where a poverty index of ≥3.5 indicates a high level of poverty [29]. Number of children in the household was 0–3 or greater than 4 [12].

It is not clear from the literature if level of education is associated with asthma after adjusting for income [16,27]. We calculated Akaike information criteria (AICs) for the ordinal and logistic regression models with and without the variable "education of the household reference person" to evaluate model fit. In both cases, model fit was improved by including the level of education in the model.

We tested for interaction effect between: "age" and "poverty index", "race" and "poverty index"; and "poverty index" and "number of children in the household." There was no significant interaction between "age" and "poverty index". We detected significant interaction (p < 0.05) for the latter two combinations and included these interaction terms in the final model.

Because *T. gondii* infection and asthma prevalence vary with age (Fig 3), and the relationship between asthma and atopy could also differ with age [2], we conducted additional ordinal regression analyses stratified by age category to check for effect modification and calculated the AIC values to compare model fit (Table 2).

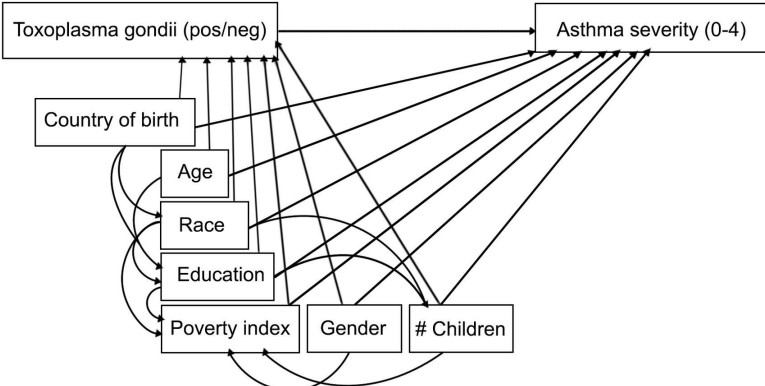

**Fig 2. Directed acyclic graph illustrating the relationships amongst the exposure variable "*Toxoplasma gondii*", the outcome variable "Asthma severity", and adjusted confounders.** *T. gondii* infection is characterized as positive (pos) or negative (neg) and Asthma severity is categorized from 0 to 4, where 0 indicates no asthma and 4 indicates the subject has asthma that caused at least one emergency room visit in the last year. Age is age in years at last birthday. Education refers to the highest level of education attained by the household reference person. Poverty index refers to the poverty index assigned to the household reference person and # Children is the number of children <18 years of age living in the household.

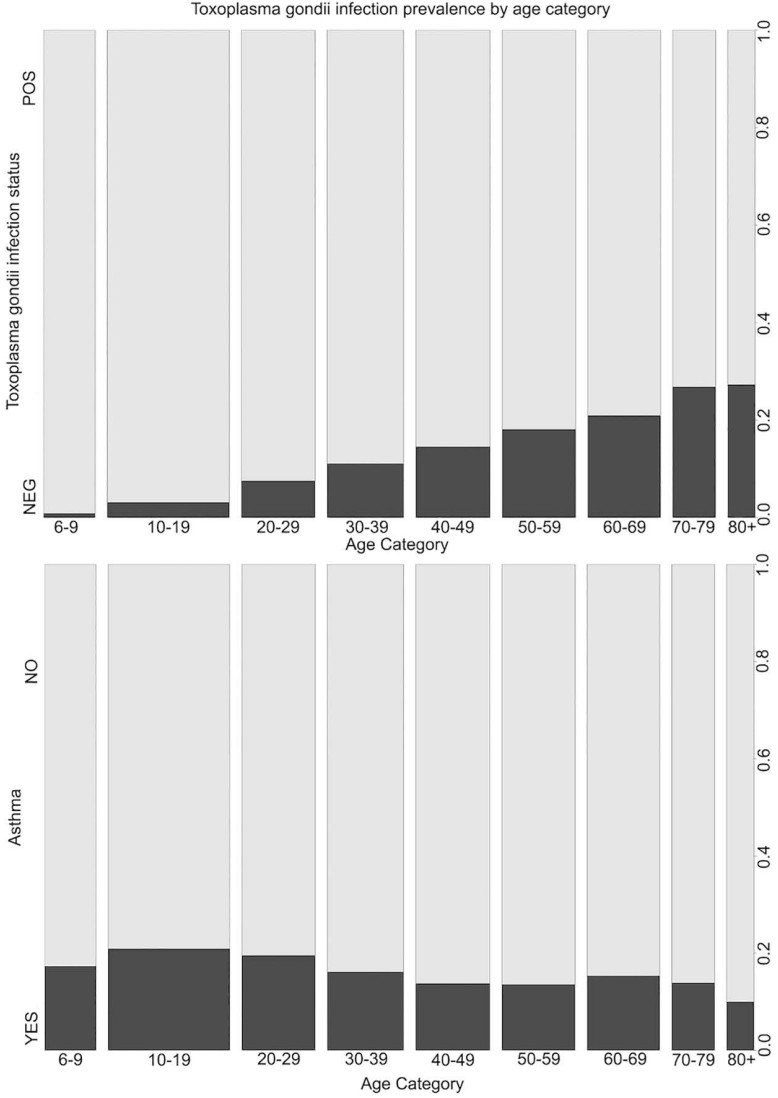

**Fig 3. *Toxoplasma gondii* and asthma prevalence by age category in the U.S. non-institutionalized civilian population age 6 years and older.** Data is from the NHANES 2011-2014 survey cycles (n = 12,620).

Analyses were performed in R (version 4.3.1) [30]. We used a design-based analysis approach to account for survey weights and complex survey design using the CRAN survey package (version 4.2−1) [31]. Descriptive statistics are presented in Table 1; reported frequencies are based on the unweighted analytic sample while the percentages account for complex survey design.

Among all subjects, country of birth was missing for 0.03%, education level was missing for 3.29%, poverty index was missing for 6.86%, and age was missing for 19.0%. We assumed the data were missing at random (MAR) and imputed the missing covariate data using five imputations [32]. All analyses were conducted on each imputed data set and results were pooled using Rubin's rules [33]. We compared results using imputed data with those obtained from a complete case analysis.

We fit an ordinal logistic regression model to estimate odds ratios (OR) and 95% confidence intervals (CI) of the unadjusted association between *T. gondii* infection and asthma severity (0–4), and the association after adjusting for all

**Table 2. Odds ratios (OR) and 95% confidence intervals (CI) for the relationship between *Toxoplasma gondii* infection status and asthma in the U.S. non-institutionalized civilian population, stratified by age category. Asthma is expressed as an ordinal categorical variable with levels of severity 0-4. Model fit is evaluated using AIC[a]. Data is from the NHANES 2011-2014 survey cycles.**

| Ordinal logistic regression[b] for asthma severity 0–4 | OR (95% CI) | AIC[a] |
|---|---|---|
| All ages | | |
| *Toxoplasma gondii* | | |
| Positive | 0.90 (0.83-0.97) | 14780 |
| **Age 6–9** | | |
| *Toxoplasma gondii* | | |
| Positive | 0.75 (0.36-1.58) | 1235 |
| **Age 10–19** | | |
| *Toxoplasma gondii* | | |
| Positive | 0.78 (0.80- 1.21) | 3742 |
| **Age 20–29** | | |
| *Toxoplasma gondii* | | |
| Positive | 0.91 (0.59-1.40) | 2019 |
| **Age 30–39** | | |
| *Toxoplasma gondii* | | |
| Positive | 1.24 (0.96-1.60) | 1828 |
| **Age 40–49** | | |
| *Toxoplasma gondii* | | |
| Positive | 0.83 (0.60-1.14) | 1560 |
| **Age 50–59** | | |
| *Toxoplasma gondii* | | |
| Positive | 0.81 (0.60-1.09) | 1524 |
| **Age 60–69** | | |
| *Toxoplasma gondii* | | |
| Positive | 1.31 (0.95-1.81) | 1836 |
| **Age 70–80+** | | |
| *Toxoplasma gondii* | | |
| Positive | 1.03 (0.86-1.22) | 1836 |

[a]Akaike information criterion. AIC does not offer a comparison of model fit across age categories because they are not identical data sets.

[b]Model incorporates complex survey design and is adjusted for gender, poverty index, race, poverty index-*race, country of birth, number of children in the household, number of children in the household*poverty index, and education of the household reference person.

confounders (Table 3). For comparison, we fit a logistic regression model to estimate the unadjusted and fully adjusted association between *T. gondii* infection and asthma as a bivariate outcome, "Yes" or "No".

## Results

Of the 12,620 subjects 6 years of age or older tested for *T. gondii* infection, 10.8% were seropositive and 89.2% were seronegative. Most subjects (83.5%) reported that they had never been told that they had asthma (level 0), while 16.5% reported varying degrees of asthma severity (levels 1–4). Asthma prevalence varied by age category (Fig 3), with the highest prevalence found in 10–29-year-olds and the lowest in subjects 80 years and older. Prevalence of *T. gondii*

**Table 3. Odds ratios (OR) and 95% confidence intervals (CI) for the unadjusted and adjusted relationship between _Toxoplasma gondii_ infection status and asthma in the U.S. non-institutionalized civilian population age 6 years and older. Asthma is expressed as an ordinal categorical variable with levels of severity 0-4 in an ordinal logistic regression model and as a bivariate yes/no variable in a logistic regression model. Data is from the NHANES 2011-2014 survey cycles[a].**

|  | Unadjusted OR (95% CI) | Adjusted[b] OR (95% CI) |
|---|---|---|
| Ordinal logistic regression for asthma severity 0–4 |  |  |
| _Toxoplasma gondii_ |  |  |
| Negative | reference | reference |
| Positive | 0.90 (0.83-0.97) | 1.00 (0.91 - 1.10) |
| Logistic regression for asthma yes/no |  |  |
| _Toxoplasma gondii_ Negative | reference | reference |
| Positive | 0.82 (0.7-0.95) | 1.02(0.85-1.21) |

[a]Survey design is accounted for in the analysis and missing covariate data is addressed using multiple imputation.

[b]Model is adjusted for gender, age, poverty index, race, poverty index*race, country of birth, number of children in the household, number of children in the household*poverty index, and education of the household reference person.

infection increased with age (Fig 3). No statistically significant difference was detected for seroprevalence of _T. gondii_ across the five levels of asthma severity ($p = 0.140$). Significant unadjusted associations between asthma and the covariates gender ($p = 0.036$), age category ($p = 0.019$), poverty index ($p = 0.003$), and race ($p = 0.024$) are presented in Table 1.

No significant relationship was detected between _T. gondii_ infection and asthma. A complete case analysis using the unadjusted ordinal regression model returned an OR of 0.89 (95% CI = 0.79–1.01), suggesting an insignificant protective effect of _T. gondii_ infection. Adjusting for all confounders, the complete case model returned an OR of 1.02 (95% CI 0.87–1.19). Using multiple imputation to account for missing covariate data, the unadjusted regression model returned an OR of 0.90 (95% CI = 0.83–0.97), suggesting a protective effect from _T. gondii_ infection, but the fully adjusted model returned an OR of 1.00 (95% CI = 0.91–1.10), confirming there was no detected association between _T. gondii_ infection and asthma (Table 3). The unadjusted and adjusted logistic regression models evaluating asthma as a bivariate outcome variable (Yes/No) yielded similar results (Table 3). The ordinal model fit the data better than the logistic model according to the AIC values.

Regression analyses stratified by age category suggested that there is some effect modification by age; the association between _T. gondii_ infection and asthma severity varied across age categories, but within each category the association remained insignificant (Table 2).

## Discussion

No significant relationship was detected between _T. gondii_ infection and asthma in U.S. residents from the 2012–2014 NHANES survey cycles when models were adjusted for key demographic factors. The NHANES data has some limitations, however. An estimated 20% of asthma cases are undiagnosed in the U.S. and therefore not captured by the NHANES [31,34], which could bias our results. Additionally, the data are cross-sectional and cannot capture temporal relationships between exposure and outcome. If _T. gondii_ exerts an effect on asthma over only a short period post-infection, our analysis might fail to detect this. Furthermore, asthma is a syndrome comprised of several diseases, not all of which are associated with atopy (4). Because _T. gondii's_ potential impact on asthma is postulated to occur via immune system modulation (6,9), the inability to distinguish atopy-related asthma from other types would likely reduce the sensitivity of our analysis.

Model fit was improved when stratified by age category, and ORs appeared to vary across age categories but remained statistically insignificant (Table 3). The non-linear relationship between asthma and age (Fig 3) might contribute to this effect.

For all adjusted regression models, no significant relationship was detected between *T. gondii* infection and asthma. These findings are consistent with two other large cross-sectional studies in Scotland and Australia that found no association between *T. gondii* seropositivity and asthma [14,35]. While our results appear to contradict several case-control studies that have found a protective effect of *T. gondii* infection on allergy symptoms and asthma, these studies did not consider confounders which could impact exposure to a diversity of microorganisms and their products, such as socio-economic status or number of siblings [36–38]. *Toxoplasma gondii* is, therefore, likely a marker for the protective effect of exposure to a diversity of microbial organisms and their products early in life, but may not have any causal effect on this phenomenon, or may not be important when considered in isolation.

## Acknowledgments

The author thanks Michael Asamoah-Boaheng for reviewing preliminary drafts of this manuscript.

## Author contributions

**Conceptualization:** Heather Anholt.

**Formal analysis:** Heather Anholt.

**Project administration:** Heather Anholt.

**Resources:** Heather Anholt.

**Visualization:** Heather Anholt.

**Writing – original draft:** Heather Anholt.

**Writing – review & editing:** Heather Anholt.

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
