## [Decision Letter · Decision Letter 0]

PONE-D-24-15660The association between Toxoplasma gondii infection and asthma in the United States: A cross-sectional survey analysis.PLOS ONE

Dear Dr. Anholt,

Thank you for submitting your manuscript to PLOS ONE. After careful consideration, we feel that it has merit but does not fully meet PLOS ONE’s publication criteria as it currently stands. Therefore, we invite you to submit a revised version of the manuscript that addresses the points raised during the review process.

We look forward to receiving your revised manuscript.

Kind regards,

Masoud Foroutan, Ph.D; Assistant Professor

Academic Editor

PLOS ONE

Journal Requirements:

1. When submitting your revision, we need you to address these additional requirements. Please ensure that your manuscript meets PLOS ONE's style requirements, including those for file naming. The PLOS ONE style templates can be found at https://journals.plos.org/plosone/s/file?id=wjVg/PLOSOne_formatting_sample_main_body.pdf and https://journals.plos.org/plosone/s/file?id=ba62/PLOSOne_formatting_sample_title_authors_affiliations.pdf

Reviewers' comments:

Reviewer's Responses to Questions

**Comments to the Author**

1. Is the manuscript technically sound, and do the data support the conclusions?

Reviewer #1: Yes

Reviewer #2: Yes

2. Has the statistical analysis been performed appropriately and rigorously? 

Reviewer #1: Yes

Reviewer #2: Yes

3. Have the authors made all data underlying the findings in their manuscript fully available?

Reviewer #1: Yes

Reviewer #2: Yes

4. Is the manuscript presented in an intelligible fashion and written in standard English?

Reviewer #1: Yes

Reviewer #2: Yes

5. Review Comments to the Author

Reviewer #1: The article was technically sound with data supporting the final conclusion. These supporting data are thoroughly explained in the results section with appropriate statistical methods and adjustments. All data are easily available, and the manuscript is very well written in intelligible fashion.

Reviewer #2: Dear Editors,

I am writing to confirm my willingness to referee Manuscript Number PONE-D-24-15660, titled “The Association Between Toxoplasma gondii Infection and Asthma in the United States: A Cross-Sectional Survey Analysis” submitted to PLOS ONE. I appreciate the opportunity to contribute to the peer review process for this manuscript.

The abstract of the manuscript outlines a study investigating the relationship between Toxoplasma gondii infection and asthma using data from the 2012-2014 American National Health and Nutrition Examination Survey (NHANES). As an immunologist with extensive expertise in the interactions between microbial influences and immune responses, I am well-positioned to evaluate the study’s methodology and analysis.

Please let me know if there are any specific aspects you would like me to focus on or if there are additional details I should be aware of during the review process.

Thank you for considering me as a reviewer for this manuscript. I look forward to contributing to the evaluation of this important research.

Overall Assessment:

In recent years, the association between allergic illnesses, particularly asthma, and numerous infectious pathogens has garnered significant attention. The hygiene concept, highlighted by the author in this publication, is a particularly intriguing subject within immunology. This study examined the correlation between T. gondii infection and asthma. This matter has not been thoroughly examined to date. Approximately 12% of the American population is infected with T. gondii, underscoring the imperative for this research. This investigation is excellent in terms of both its design and Metodology. Additionally, ethical considerations are assessed. Ultimately, with little revisions and modifications, the publication of this content can be quite beneficial and will, conversely, captivate readers' interest. Below are few recommendations to enhance the article.

1. Validity of the Research

• The research delineates a precise methodology for participant selection, incorporating explicit inclusion and exclusion criteria. This guarantees that the study population is suitable and representative of the research aims. The sample size is adequately determined and sufficient to produce accurate results, hence reducing the likelihood of statistical error.

• The statistical approaches utilized in the analysis are both conventional and suitable for the data set. The proper application of these methodologies further enhances the validity and robustness of the study's conclusions.

2. Clarity of Reporting

-Abstract: The abstract of the study would benefit from a more comprehensive reference to the principal findings. Emphasizing these pivotal conclusions in the abstract will furnish readers with a more lucid comprehension of the study's importance and findings from the beginning. Highlighting the principal findings in this part will augment the abstract’s efficacy in communicating the study’s significance and relevance.

- Introduction: It is advised that one paragraph be devoted to exploring the influence that T. gondii has on the immune system in the introduction. Particular focus should be placed on the important components that contribute to the emergence of an allergic response.

- Methods: The methods section of the study is thoroughly described, ensuring that the procedures can be effectively replicated. This detailed explanation of the methodologies employed not only facilitates reproducibility but also underscores the study's commitment to due diligence. By providing a comprehensive account of the techniques and tools used, the study upholds rigorous standards of research practice.

- Results: The results of the study are clearly presented both in the text and through figures and tables. These presentations accurately reflect the detailed analysis conducted, ensuring that the findings are comprehensible and well-supported by the data.

- Discussion: The discussion section provides a thorough interpretation of the study's findings, situating them within the broader context of existing research. It effectively explores the implications of the results, acknowledges the study's limitations, and suggests avenues for future research. The analysis is well-structured and integrates the findings with relevant literature, offering valuable insights into the topic under investigation.

3. Novelty and Impact

This research makes a significant contribution to the field of epidemiology and public health by advancing the understanding of the relationship between asthma and Toxoplasma gondii. The study's findings provide new insights that enhance the current knowledge base and influence contemporary perspectives on these health issues. By clarifying the interplay between these factors, the research offers valuable implications for both theoretical understanding and practical applications in public health.

4. Suggestions for Improvement

It is strongly recommended that the article undergo a thorough review for grammar and writing. Ensuring grammatical accuracy and refining the writing style will enhance the clarity and professionalism of the manuscript, contributing to a more polished and effective presentation of the research.

5. Recommendation:

- Minor revisions required

6. PLOS authors have the option to publish the peer review history of their article (what does this mean? ). If published, this will include your full peer review and any attached files.

**Do you want your identity to be public for this peer review?** For information about this choice, including consent withdrawal, please see our Privacy Policy .

Reviewer #1: **Yes: ** Diego Rosado

Reviewer #2: No

---

## [Author Response · Author response to Decision Letter 1]

4 May 2025

Record of changes to the manuscript

- I have changed the level 1 headings to sentence case.

- I have renamed the figure tiff files.

- I have changed the citation style from Vancouver to PLOS ONE.

- I have made tracked formatting changed to the manuscript in accordance with:

and

Reviewer 2: Clarity of Reporting

Abstract: The abstract of the study would benefit from a more comprehensive reference to the principal findings. Emphasizing these pivotal conclusions in the abstract will furnish readers with a more lucid comprehension of the study's importance and findings from the beginning. Highlighting the principal findings in this part will augment the abstract’s efficacy in communicating the study’s significance and relevance.

lines 22-30: The following text has been added to the Abstract.

Of the 12,620 subjects tested for T. gondii infection, 89.2% were seronegative and 10.8% seropositive. No asthma was reported by 83.5% of subjects, while 16.5% reported varying degrees of asthma severity. We detected no significant association between T. gondii infection and asthma. While the unadjusted regression model suggested a small protective effect of T. gondii on asthma (OR = 0.90; 95% CI = 0.83-0.97), no effect was detected when the model was adjusted for key demographic factors (OR = 1.00, 95% CI = 0.91-1.10). While T. gondii may be a marker for the protective effect of exposure to a diversity of microbial organisms early in life, it has no apparent causal effect on asthma, or it may not be significant when considered in isolation.

Reviewer 2: Introduction

It is advised that one paragraph be devoted to exploring the influence that T. gondii has on the immune system in the introduction. Focus should be placed on the important components that contribute to the emergence of an allergic response.

lines 48-63: the last paragraph of the Introduction now reads as follows:

Toxoplasma gondii is an obligate intracellular gastro-intestinal protozoan, and nearly ubiquitous microorganism, whose definitive host is cats. People become infected with T. gondii through indirect fecal-oral transmission or consumption of undercooked meat. About 12% of the U.S. population is infected, and in some countries, prevalence is as high as 60% [7,8]. Infection with T. gondii induces a strong cell-mediated immune response, with the production of IFN-g cytokines by polarized T helper type 1 cells (Th1) in the early stages of infection [9,10]. Conversely, the pathophysiology of asthma is characterized by the development of hyperreactivity due to cytokines produced by T helper type 2 cells (Th2). Infection with T. gondii may exert a protective effect on the development of antigen-induced airway inflammation through high concentrations of Th1 cytokines and a reduction of allergen-specific Th2-associated cytokines associated with the T. gondii immune response [11]. Indeed, Fenoy et al. (2009) showed that T. gondii infection substantively blocked the development of antigen-induced airway inflammation in adult laboratory (BALB/c) mice [9]. Despite recognized impacts on the mammalian immune response, however, the relationship between T. gondii infection and asthma has not been well investigated [6,8]. In the current study, we quantify the association between T. gondii infection and asthma in the U.S. population.

Reviewer 2: It is strongly recommended that the article undergo a thorough review for grammar and writing.

- The manuscript has been reviewed by a Wiley associate editor and the following changed have been made in response to their comments.

- Tables 2 and 3 appeared in reverse order. They have been renumbered.

- The following text has been added to lines 169-170 (Table 2):

o “AIC does not offer a comparison of model fit across age categories because they are not identical data sets.”

- lines 231-233:

o This paragraph has been edited to avoid direct comparison of AIC values across data sets. It now reads:

Regression analyses stratified by age category suggested that there is some effect modification by age; the association between T. gondii infection and asthma severity varied across age categories, but within each category the association remained insignificant (Table 2).

- lines 235-237

o the main result is reiterated in the first sentence of the Discussion as follows:

No significant relationship was detected between T. gondii infection and asthma in U.S. residents from the 2012-2014 NHANES survey cycles when models were adjusted for key demographic factors.

- line 253: “might” has been changed to “would likely”.

---

## [Decision Letter · Decision Letter 1]

The association between Toxoplasma gondii infection and asthma in the United States: A cross-sectional survey analysis.

PONE-D-24-15660R1

Dear Dr. Anholt,

We’re pleased to inform you that your manuscript has been judged scientifically suitable for publication and will be formally accepted for publication once it meets all outstanding technical requirements.

Kind regards,

Masoud Foroutan, Ph.D; Assistant Professor

Academic Editor

PLOS ONE

Additional Editor Comments (optional):

Reviewers' comments:

Reviewer's Responses to Questions

**Comments to the Author**

1. If the authors have adequately addressed your comments raised in a previous round of review and you feel that this manuscript is now acceptable for publication, you may indicate that here to bypass the “Comments to the Author” section, enter your conflict of interest statement in the “Confidential to Editor” section, and submit your "Accept" recommendation.

Reviewer #2: All comments have been addressed

2. Is the manuscript technically sound, and do the data support the conclusions?

Reviewer #2: Yes

3. Has the statistical analysis been performed appropriately and rigorously? 

Reviewer #2: Yes

4. Have the authors made all data underlying the findings in their manuscript fully available?

Reviewer #2: Yes

5. Is the manuscript presented in an intelligible fashion and written in standard English?

Reviewer #2: Yes

6. Review Comments to the Author

Reviewer #2: Dear Author,

I am writing to confirm my willingness to referee Manuscript Number PONE-D-24-15660R1, titled “The Association Between Toxoplasma gondii Infection and Asthma in the United States: A Cross-Sectional Survey Analysis” submitted to PLOS ONE. I appreciate the opportunity to contribute to the peer review process for this manuscript.

All reviewers’ comments have been carefully addressed, and the necessary revisions have been made accordingly.

I believe that the manuscript has been significantly improved and is now suitable for publication.

Please let me know if there are any specific aspects you would like me to focus on or if there are additional details I should be aware of during the review process.

Thank you for considering me as a reviewer for this manuscript. I look forward to contributing to the evaluation of this important research.

7. PLOS authors have the option to publish the peer review history of their article (what does this mean? ). If published, this will include your full peer review and any attached files.

**Do you want your identity to be public for this peer review?** For information about this choice, including consent withdrawal, please see our Privacy Policy .

Reviewer #2: No
